# Planning Paths through Occlusions in Urban Environments

**Yutao Han**[*]
OPPO US Research Center
{yutao.han}@innopeaktech.com

**Youya Xia**[*]
Cornell University
{yx454}@cornell.edu

**Guo-Jun Qi**
OPPO US Research Center
{guojun.qi}@innopeaktech.com

**Mark Campbell**
Cornell University
{mc288}@cornell.edu

**Abstract:** This paper presents a novel framework for planning in unknown and occluded urban spaces. We specifically focus on turns and intersections where occlusions significantly impact navigability. Our approach uses an inpainting model to fill in a sparse, occluded, semantic lidar point cloud and plans dynamically feasible paths for a vehicle to traverse through the open and inpainted spaces. We demonstrate our approach using a car's lidar data with real-time occlusions, and show that by inpainting occluded areas, we can plan longer paths, with more turn options compared to without inpainting; in addition, our approach more closely follows paths derived from a planner with no occlusions (called the ground truth) compared to other state of the art approaches.

**Code**: https://github.com/genplanning/generative_planning

**Keywords:** Navigation, Occluded Environments, Semantic Scene Understanding

## 1  Introduction

Planning in environments with unknown spaces is a challenging topic in robotics, as they can limit the speed of travel and decision making in real-time. Unknown spaces occur from occluding objects, such as cars in the road, buildings, trees or fences, or from limitations in sensor range and resolution. The number and extent of these unknown spaces increase dramatically in cluttered environments, such as in an airport or in an urban city. Traditionally, path planners in these types of environments typically either plan only in the known spaces (which limits speed and navigability) or assumes unknown space is free (which increases the chances of varied maneuvering and collisions).

Consider an example of a car turning at an upcoming intersection, but there are pedestrians on the corner and a truck is in the intersection, such that space beyond the truck/pedestrians is unknown. If a path planner only plans in known spaces, the planner will not have the range to plan a path through the intersection. If the planner assumes unknown space is free, it could potentially run into dead ends. Similar examples exist regularly in urban environments. Comparatively, a human driving in such an environment makes predictions about what lies beyond the occluding objects in order to navigate more smoothly. If unable to predict what is behind occluding objects, drivers will slow down significantly before advancing further.

Our approach, shown in Figure 1, addresses this problem by filling in unknown spaces using image inpainting techniques. We define unknown spaces as regions of previously unmapped environments occluded from sensor measurements by obstacles such as buildings. Our work predicts the underlying static structure in these unknown spaces, for example, a turn or intersection in the road for more informed path planning. First, a sensor measurement is projected to the bird's eye view (BEV) in order to reveal occluded spaces. Next, a data-driven image inpainting model fills in the occluded spaces. Finally, the inpainted map is used for planning a dynamically feasible path. We specifically focus on turns and intersections where occlusions significantly impact navigation.

---

[*]Equal contribution

6th Conference on Robot Learning (CoRL 2022), Auckland, New Zealand.

| BEV Lidar Scan | Inpainted Map | Skeletonized Road | Planned Path |
| --- | --- | --- | --- |

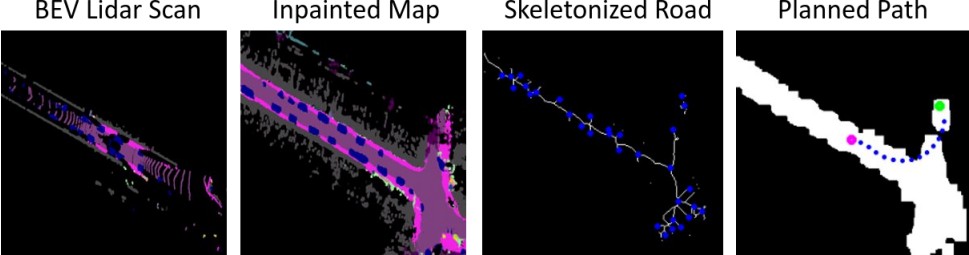

Figure 1: (L→R) (1) Initial lidar scan rendered from a BEV perspective annotated with semantic labels (purple = road). (2) Output of an inpainting model used on the initial lidar scan; unknown pixels are filled in with semantic labels. (3) The skeletonized road; blue dots represent skeletonized nodes where multiple edges connect. (4) The planned path through white pixels denoting traversable road; the pink dot is the initial pose, the green dot is the goal, and blue dots are the path sequence.

The main contributions of this paper are:

- a predictive inpainting model for navigation in urban outdoor environments which fills in unknown spaces, extending the perception range allowing for smoother and faster navigation, especially for turns and intersections in cluttered environments. The predictive model is easily incorporated into existing path planning methods.

- a modified loss to the Pix2Pix [1] network tailored to the occlusion inpainting task.

- a dataset for training models of occluded turns and intersections for filling in unknown spaces in urban environments.

The proposed framework is compared against a planner which does not use any inpainting to fill in occluded spaces. Experimental results demonstrate the novel framework is highly effective at (1) identifying turns and intersections through occluding objects, (2) generating a traversability graph which closely matches the ground truth (i.e. path assuming no occlusions), and (3) extending the range of dynamically feasible trajectory planning in occluded environments.

## 2   Related Works

Path planning in known environments is a well studied problem in robotics. Grid-based map representations are generally used and many optimal algorithms exist (A* [2], D* [3]). Unknown spaces (due to occlusions) are generally modeled as free or ignored [4]. Approaches that model the unknown space as free must frequently replan online due to discrepancies between what is actually in the unknown space and the optimistic assumption that unknown space is free [5].

Ref. [6] shows the range of a path planner in outdoor urban environments can be extended by using semantic segmentations. This approach demonstrates that by extending the trajectory length, the robot navigates faster and traverses smoother paths compared to a traditional metric grid planner. This approach however does not address planning in occluded spaces.

**Data-driven predictive modeling** of unknown spaces has been studied in recent work. The previous work most similar for outdoor environments [5] uses image inpainting to fill in unknown spaces from sensor measurements projected to a BEV. This work requires explicit labeling of the map pixels which must be inpainted. In our approach, we do not explicitly label which pixels must be inpainted as the pixel inpainting is completely data-driven. Our approach also considers the dynamics of the vehicle, while [5] does not. Refs. [7, 8] use predictive modeling of unknown spaces for planning, but their scope is restricted to indoor environments.

[9, 10, 11] use image inpainting to fill in foreground objects with background classes. However, these approaches do not directly integrate with path planning.

**Generative Adversarial Networks** (GANs) [12] are a powerful tool used for a variety of tasks, such as natural language processing [13], super resolution [14], and image translation [15]. Generally, GANs aim to model the underlying distribution lying in the target data domain. Traditional GANs for image inpainting tasks can be divided into two categories. The first are GANs targeted for paired image-to-image translation such as Pix2Pix [1] and Pix2PixHD [16]. The other type of GANs,

considered to be state-of-the-art image inpainting models [17, 18], use free-form or rectangular masks as inpainting signals.

## 3 Technical Approach

Figure 1 shows the pipeline for our proposed approach. First, a semantic lidar point cloud derived from a lidar scan is created. The point cloud is transformed to a BEV and an inpainting model fills in the unknown, occluded pixels. Next, the traversable road pixels are skeletonized into a graph with waypoints. Finally, a dynamically feasible path is planned from the vehicle pose to a given goal.

### 3.1 Dataset Generation

Our dataset is generated based on the KITTI-360 dataset [19]. In the KITTI-360 dataset, a vehicle is driven around Karlsruhe, Germany, and sensor data is collected and annotated for 73.7 km. We specifically focus on semantic lidar data in this work. Lidar provides more reliable depth than stereo cameras, but still suffers from issues with occluding obstacles blocking out unknown space. KITTI-360 provides semantic annotations (19 classes) for the *aggregated* lidar point clouds for each route.

For our work, we transform point clouds to a bird's eye view (BEV) because BEV allows for clear observation of occluded spaces. We project semantic annotations from the fully aggregated lidar point clouds to each individual lidar scan. Then, we render both the aggregated point clouds and the individual semantic lidar scans from a BEV to complete our dataset. We remove points with class label vegetation since vegetation can obscure the underlying road and sidewalks, and also unknown spaces underneath in BEV images. Since the occlusions are already present in the point cloud, removing the vegetation points does not impact our inpainting results. We have rendered 22698 frames for three driving sequences.

### 3.2 Image Inpainting

In the lidar measurements there are unknown spaces such as shown in Figure 3 (top left). The orange oval shows semantic annotations for a region that is occluded in Figure 3 (top second). These occluded spaces are a function of obstacles such as buildings and fences, and also sensor resolution. To predict the semantics in the unknown spaces, we employ a paired generation model to recover the lost semantic information.

#### 3.2.1 Baseline Paired Generation Algorithm

For translating an image $x^{(n)}$ in source domain $\mathcal{X}$ ($\mathcal{X}$ is defined as the 2D lidar space) to an image $y^{(n)}$ in target domain $\mathcal{Y}$ ($\mathcal{Y}$ is defined as the 2D ground truth semantic space), we first introduce a baseline paired generation algorithm, inspired by [1]. Isola et al. [1] assumes the training data are perfectly paired images, and an L1 loss in the pixel space compares $x^{(n)}$ and $y^{(n)}$:

$$\mathcal{L}_1(G, X, Y) = \frac{1}{N} \sum_n \frac{1}{K^{(n)}} \sum_{i,j} \|G(x^{(n)})_{i,j} - y^{(n)}_{i,j}\|_1. \tag{1}$$

where $G(x^{(n)})_{i,j}$ and $y^{(n)}_{i,j}$ are pixel values at the $(i,j)$ location and $K^{(n)}$ is the number of pixels in image $y^{(n)}$. $N$ refers to the number of images in the training set. Additionally, [1] further employs a standard GAN loss to restore a realistic image. We ask the reader to refer to [12] for the implementation of a standard GAN.

#### 3.2.2 Implementation

To ensure the paired generation model restores most of the semantics lost in lidar space, we propose a modification of the Pix2Pix network. First of all, we adopt a coarse-to-fine generator $G$ and a multi-scale discriminator $D$ from Pix2PixHD [16] which is the state-of-the-art paired generation model.

### 3.2.3 The inpainting-targeted L1 loss

Although the generative model restores some unknown semantics in lidar space, the original semantics sometimes can get lost during the feature extraction process (see Figure 2) due to the sparsity of extractable features. To ensure the input semantics are preserved during the feature extraction process, we propose an inpainting-targeted L1 loss. Specifically, for an input image $x^n \in \mathcal{X}$ and the output image $G(x^n \in \mathcal{Y})$, the loss is defined as:

$$\mathcal{L}_1^\star(G, X) = \frac{1}{N} \sum_n \frac{1}{K^{x_n}} \sum_{(i,j) \in x^{*,n}} \|G(x^n)_{i,j} - x_{i,j}^n\|_1. \tag{2}$$

where $K^{x_n}$ refers to the number of non-zero pixels in the input lidar image $x^n$ and $(i,j) \in x^{*,n}$ refers to the pixel location $(i,j)$ in the nonzero set $x^{*,n}$ of the original input lidar image. Thus, from the loss function definition, the inpainting-targeted L1 loss aims to ensure the nonzero semantics lying in the original input image $x^n$ still exist in the generated image $G(x^n)$. Figure 2 shows an ablation study of $\mathcal{L}_1^\star(G, X)$.

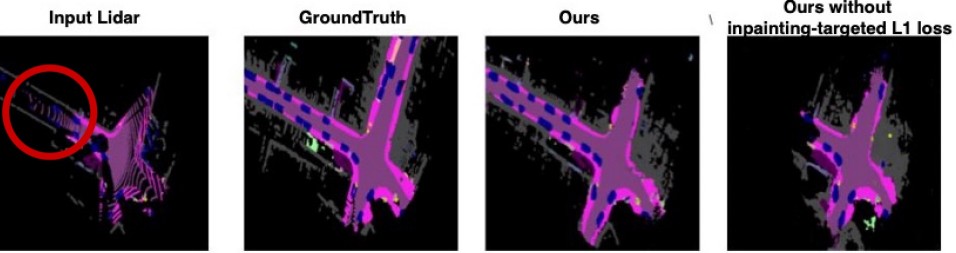

Figure 2: Comparison of the inpainting performance on our model with and without inpainting-targeted L1 loss $\mathcal{L}_1^\star(G, X)$ (Sec. 3.2.3). L→R: (1) input lidar, (2) GT semantics, inference results from our model (3) with and (4) without $\mathcal{L}_1^\star(G, X)$. Sparse semantics in the faraway region of the road in the original lidar map (red circle) are predicted from our model with $\mathcal{L}_1^\star(G, X)$, but not without.

### 3.2.4 Ranking-Based Loss to Overcome Large Stochastic Variations

The inpainting-targeted L1 loss effectively compensates for the semantics lost during the feature extraction stage. However, restoration of small details, especially semantics lying in the intersection areas are not solved effectively by the loss terms described above.

To address this, we employ a loss function focused on local features (*i.e.*, image patches) [20]. Patches of the generated and target images at the same position should capture similar structures and content, such that their similarity is larger than patches at different positions. We realize this idea using a loss similar to the patchNCE loss [21]. Let $H$ be a feature extractor and $H_s$ be the feature at the spatial location $s$ on the feature map. Our usage of the patchNCE loss is defined as follows

$$\sum_s \ell\left(H_s(G(x)), H_s(y), \{H_{s'}(y)|s' \neq s\}\right), \tag{3}$$

$$\text{where } \ell(v, v^+, \{v_{s'}^-\}) = -\log\left[\frac{\exp(v \cdot v^+/\tau)}{\exp(v \cdot v^+/\tau) + \sum_{s'} \exp(v \cdot v_{s'}^-/\tau)}\right]. \tag{4}$$

Here, $\tau$ is a temperature constant scalar. Through minimizing this loss with respect to $G$, we encourage $\mathbf{H_s(G(x))}$ **and** $\mathbf{H_s(y)}$ to have higher inner product similarity than $\mathbf{H_s(G(x))}$ **and other patches of y**. By applying this loss to all the training pairs, we have

$$\mathcal{L}_{\text{patchNCE}}(G, H, X, Y) = \frac{1}{N} \sum_s \ell\left(H_s(G(x^{(n)})), H_s(y^{(n)}), \{H_{s'}(y^{(n)})|s' \neq s\}\right). \tag{5}$$

In our implementation, we follow [21] to extract features from multiple layers.

Table 1: Dataset splits (# of images) for image inpainting.

| Route | Train Set | Val Set | Test Set |
|-------|-----------|---------|----------|
| Route 0 | 8386 | 1048 | 1049 |
| Route 2 | 8982 | 1123 | 1122 |
| Route 3 | 790 | 99 | 99 |

### 3.2.5  Final Training Objective

Combining the loss terms introduced in Sections 3.2.3 and 3.2.4, our training objective is

$$\mathcal{L}_{\text{GAN}}(G, D, X, Y) + \mathcal{L}_{\text{patchNCE}}(G, H, X, Y) + \mathcal{L}_1^{\star}(G, X). \tag{6}$$

Specifically, we apply mini-batch stochastic gradient descent and learn $G$ and $H$ to minimize the objective while learning $D$ to maximize the objective.

### 3.3  Path Planning

Previous work [5] uses an inpainting model to predict occluded spaces in order to plan more informed paths. However, [5] does not account for vehicle dynamics and uses an A* planner to plan with the pixels of the BEV image as possible states. In our paper, we use a hybrid A* planner [22], which plans optimal and dynamically feasible paths.

Figure 3 shows our planning pipeline. First we take as input the BEV map of either the original lidar scan (OL), inpainted image map (IM), or GT map (Figure 3 top row). Next a mask is fit to the road pixels, since the vehicle only navigates on the road. We dilate and then erode the road mask in order to connect the individual road points from the lidar scan. This allows us to visualize a fully connected road. Zhang's method [23] is used to skeletonize the road. The intersections of the skeleton define waypoints $w$ for the planner to navigate (Figure 3 middle row).

Given the final goal location, which is a point all the way around a turn at an intersection (Figure 3 top row, green dot), the closest waypoint $w$ to the goal is used as a local goal to plan to. A hybrid A* planner plans a dynamically feasible path from the vehicle pose to the goal. The planner states are $[px, py, \theta]$, where $px$ and $py$ are pixelwise coordinates and $\theta$ is orientation in the world frame.

We specifically focus on predictive modeling of the underlying static structures in the unknown spaces, such as the road. Our inpainting model can be easily combined with an existing full planning stack by using it to generate a predictive model given a sensor measurement. We present a high-level planner here, with evaluation in the experimental section, and assume an off-the-shelf low-level obstacle/collision avoidance module is used in addition to our planner.

## 4  Experimental Evaluation

### 4.1  Network Training

To train the network, described in section 3.2, we generate 2D lidar maps and the corresponding semantic maps using three traversals (sequences 0, 2, and 3) in the KITTI-360 dataset [19]. Table 1 shows the number of images in the training and test sets. The train and test sets do not contain overlapping locations. We train our model for 200 epochs. During training, the initial learning rate $lr = 0.0002$ for the first 100 epochs. For the remaining 100 epochs, the $lr$ linearly decreases to zero. We use the Adam optimizer [24] for both the generator and the discriminator ($\beta_1 = 0.5$ and $\beta_2 = 0.999$). The model is trained on one NVIDIA RTX3090 GPU. For the baselines in section 4.5, we use the same dataset split and network parameters ($lr$, Adam optimizer, and training epochs).

### 4.2  Dataset Description

We evaluate our framework by testing performance on the turns and intersections of the test set. We identify regions where the vehicle makes a turn and evaluate our method on those regions. For each turn, the dataset frames corresponding to it are selected from just before the turn is seen in the ground truth (GT) to when the vehicle reaches a point where the turn is no longer feasible. This allows us to fully evaluate the effect of the inpainting model on planning for the turns and intersections, which

are the most difficult regions to plan through. For example, for the first turn, fifty frames are used for evaluation. In each frame a path is planned from the current vehicle location to the selected goal point from the skeletonized nodes. The paths consist of a set of nodes as described in section 3.3.

## 4.3 Evaluation metrics

**Frechet distance [25]** is a popular metric for evaluating the similarity between two curves. It is defined as the shortest pairwise distance between the two curves able to traverse both curves completely. Since the planned paths are 2D curved shapes, we choose the Frechet distance [25] to measure the similarity between the planned paths using our model and the GT paths.

**Average Angle Difference** evaluates if the inpainted map (IM) allows the vehicle to make more informed decisions around turns by comparing the orientation ($\theta$) in the world frame of the planned trajectories for the original lidar (OL) and IM compared to the GT map. For each trajectory $\mathcal{T}$ (from either OL or IM) we compare ($\theta$) of each path node to ($\theta$) of the closest node from the GT trajectory. The average of the angle differences for each trajectory evaluates how accurately the planned trajectory follows the GT trajectory. The Average Angle Difference (AAD) calculation is introduced

$$AAD(in, gt) = \frac{1}{n} \sum_{i \in \mathcal{T}} \left| \theta_{in,i} - \theta_{gt,i} \right| \tag{7}$$

where $\theta_{in,i}$ refers to the $ith$ node of the OL or the IM and $\theta_{gt,i}$ refers to the $ith$ node of the GT map. $n$ refers to the number of planned trajectory nodes in the IM or the OL.

**Accuracy of Major Branch Prediction per Skeleton** compares the skeletonized road graphs (Figure 3 middle row) for OL and IM to skeletonized graphs for GT maps. Comparing the skeletonized graphs allows for evaluation of the improvement the IMs have on the range of planner and how closely the predictions from the inpainting model match the GT map from a practical planning perspective. If the generated skeleton provides more planning hypotheses than the OL, it allows for planning with a wider range of start and end locations. The number of road branches (defined as the number of different turns that can be taken at an intersection) in the skeleton is the metric used to indicate the possible range of planning hypotheses. To compute the road branch prediction accuracy we calculate the percentage $\frac{N_{im}}{N_{gt}} \times 100$ where $N_{im}$ and $N_{gt}$ are the number of branches in the IM and GT skeletons respectively. We perform the same evaluation for OL.

**Path Length** directly reflects if the planner plans to the desired goal location in the GT map precisely. We evaluate the accuracy of the planned trajectory by comparing its length with the GT trajectory. The length of the trajectories are calculated using the L2 distances between path nodes. We compare the length of the IM trajectory $\mathcal{L}_{im}$ with the length of the GT trajectory $\mathcal{L}_{gt}$ using the fraction $\frac{\mathcal{L}_{im}}{\mathcal{L}_{gt}}$. We perform the same evaluation for the length of the trajectory planned using the OL.

**Planning Ahead Frames** measures how far in advance a turn is detected. If a map is sparse and limited by occlusions around turns, the planner cannot predict the turning behaviour because it does not realize the existence of a road turn. Thus, to evaluate how far ahead a turn is detected, we count how many frames before the turn or intersection the planner plans turning behavior.

## 4.4 Qualitative Evaluation

We show the evaluation results qualitatively in Figure 3. We first evaluate the top row for the semantic inpainting results. On the GT map (top second), the orange oval indicates a region occluded from the original lidar (OL) scan (top left). This is because the road section in the orange oval occurs after a turn so buildings and obstacles in the scene occlude the road after the turn. Our inpainting model (top middle) is able to predict the road's existence after the turn and accurately fill out the rest of the intersection pixels occluded in the OL scan.

The middle row compares the skeletonized road maps for our planner. Because the inpainting model fills in the occluded regions of the map around the turns, the skeletonized map for the inpainted map (IM) is much more similar in its road branching structure to the GT than the OL.

The bottom row compares the planned paths from hybrid A*. In the OL scan (bottom left), the planner is unable to see the turn highlighted by the orange oval (top second) and so the planner can

only plan a path going straight forwards. In the IM (bottom middle), the planner is able to predict the upcoming turn due to the inpainting model so it plans a path around the turn that matches the GT path.

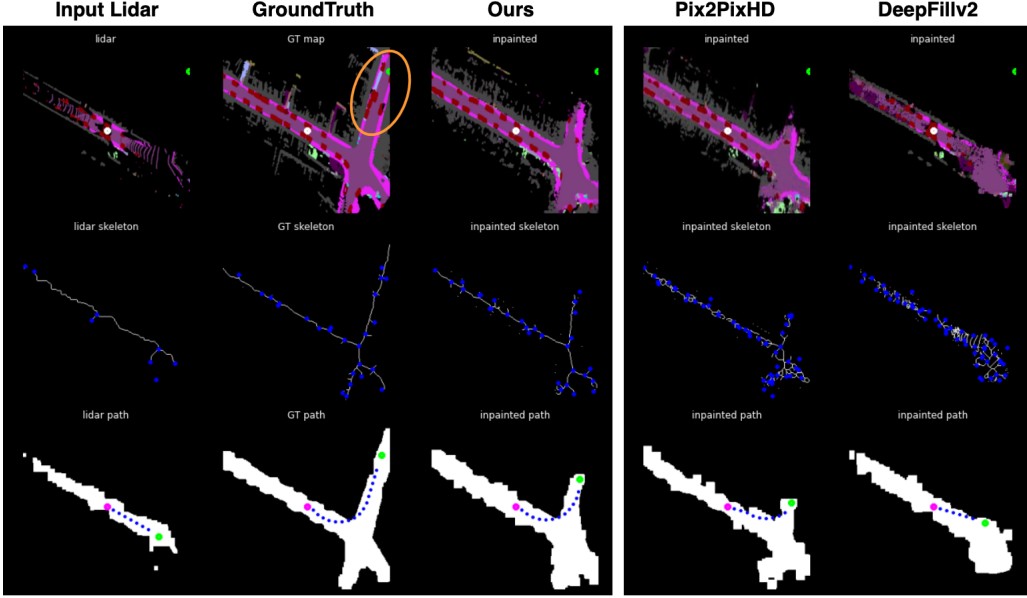

Figure 3: Comparison of inpainting and planner performance with our model and baselines: Pix2PixHD [16], DeepFillv2 [18] and ground truth (GT). Columns: (Leftmost) original lidar (OL), (Second) ground truth (GT), (Third) Ours, (Fourth) Pix2PixHD [16] and (Rightmost) Deep-Fillv2 [18] Rows: (Top) the semantic point cloud; the white dot is the vehicle pose and the green dot is the end point. The orange oval on the the top row of the GT semantic map highlights a section of the road that is occluded from the OL. (Middle) The skeletonized road with waypoints (blue dots), (Bottom) the planned path; the start is in pink, the goal is in green, and the planned nodes are in blue.

## 4.5 Quantitative Evaluation

To justify our model for the planning-under-occlusions task, we choose three different state-of-the-art inpainting networks (Pix2Pix [1], Pix2PixHD [16] and DeepFillv2 [18]) to conduct baseline comparisons. We also compare our model with the evaluation results we obtain from original lidar maps (OL) to show that generative network's predictive capabilities are an improvement for path planning compared to the OL. In addition we split our data into easy and hard sets. The easy set includes straight roads which are simple to navigate even with occluding objects and the hard set includes turns and intersections which are difficult to navigate with occluding objects. This allows for evaluation for how our framework performs in environments with different levels of difficulty (Table 3).

The evaluation results shown in Table 2 demonstrate our inpainting model outperforms the OL scans and the baselines for the path planning task. The worst performer across all metrics is the path planner on the OL scans, which means predicting semantics in unknown and occluded spaces is useful for path planning. Our model achieves the lowest Frechet Distance (8.38) and the lowest Average Angle Difference (6.73) which means our model plans the most similar paths to the GT. Using the GT map as reference, our model also achieves the highest accuracy of major branch prediction (93.01%) and path length (82.97%), which shows our model generates a road network that is the most accurate compared to the ground truth.

The evaluation results in Table 3, where our test set splits into hard and easy subsets, demonstrate the inpainting model improves performance more for complex scenes. For the Frechet Distance, our model improves over the OL by 14.12 and 12.49 (pixels) for hard and easy data respectively. For Average Angle Difference the improvement is 6.05 and 4.35 (°). For accuracy of branch prediction, improvement is 31.62 and 18.91 (%). We especially note that for frames planned ahead the improvement is the most noticeable at 19 and 1 frame(s) demonstrating our framework is especially

Table 2: **Task planning evaluation on baselines and our model using the described metrics**. *Rows* are models and *columns* are metrics. The best result for each column is **bold**.

| Model \Metric | Frechet distance (pixel) | Average Angle Difference (°) | Accuracy of Major Branch Prediction (%) | Frame Planned Ahead (#) | Path length (%) | Inference Time(FPS) |
|---|---|---|---|---|---|---|
| OL | 20.80 | 12.81 | 62.47 | 32.25 | 65.21 | N/A |
| DeepFillv2 [18] | 18.58 | 12.73 | 63.45 | 35.50 | 71.90 | 14 |
| Pix2Pix [1] | 12.09 | 10.33 | 75.40 | 42.50 | 76.63 | 16 |
| Pix2PixHD [16] | 10.06 | 9.67 | 77.88 | 47.25 | 77.02 | **33** |
| Ours | **8.38** | **6.73** | **93.01** | **52.25** | **82.97** | **33** |

Table 3: **Task planning evaluation on data divided into hard and easy subsets**. *Rows* are models and *columns* are metrics. Entries are split into (hard/easy) results. The best result for each column is **bold**.

| Model \Metric | Frechet distance ( hard / easy) (pixel) | Average Angle Difference (hard / easy) (°) | Accuracy of Major Branch Prediction (hard / easy) (%) | Frame Planned Ahead (hard / easy) (#) | Path length (hard / easy) (%) |
|---|---|---|---|---|---|
| OL | 22.67 / 17.82 | 14.90 / 7.74 | 58.44 / 70.25 | 19.75 / 12.5 | 62.67 / 93.9 |
| DeepFillv2 [18] | 18.91 / 15.17 | 12.89 / 6.76 | 61.56 / 79.40 | 23.00 / 12.5 | 69.92 / 94.81 |
| Pix2Pix [1] | 11.13 / 9.24 | 11.69 / 6.95 | 76.10 / 76.87 | 29.00 / 13.5 | 72.63 / 95.84 |
| Pix2PixHD [16] | 10.51 / 8.43 | 10.44 / 6.38 | 77.66 / 79.40 | 33.75 / 13.5 | 73.38 / 95.79 |
| Ours | **8.55 / 5.33** | **8.85 / 3.39** | **90.06 / 89.16** | **38.75 / 13.5** | **83.75 / 96.63** |

beneficial for more complex navigation scenarios such as turns and intersections. For path length, the improvement difference from hard and easy data is also large at 21.08 and 2.73 (%) respectively, demonstrating that in complex scenarios the inpainting model makes a large difference compared to the OL.

We list the inference time tested on one NVIDIA RTX3090 in the last column, which shows our model achieves acceptable inference speed (33 FPS) on available computation resources.

## 5 Limitations

Our paper assumes pose of the vehicle is known and accurate. The community has a large body of work addressing the SLAM problem which can be used for localization [26]. For our experiments we assume semantic segmentations of raw lidar point clouds can be generated in real time. Future work can use models such as [27] to generate real time semantic lidar segmentations. In addition, we assume that during online deployment, the data will be from the same domain as the training data. While this is a reasonable assumption for urban environments given the large amount of available data, in the future our framework can be extended to scenarios where online and training data are from varied domains by using domain adaptation techniques [28, 29].

## 6 Conclusion

This paper presents a novel framework for planning around unknown spaces in occluded, outdoor, urban environments. We use a data-driven inpainting model to fill in occluded regions and plan dynamically feasible paths given the model predictions. We introduce a modified loss to the Pix2Pix network and also render a dataset for the planning-through-occlusions problem. We specifically focus on turns and intersections for our evaluation as these are the regions where navigation is most heavily effected by occlusions. Experiments validate that our framework and model allow for more informed navigation in occluded spaces (especially turns), and show our planner plans paths much closer to the ground truth compared to the original sensor measurement.

**Acknowledgments**

We appreciate the reviewers for their comments and feedback. This work is funded by the ONR under the grant N00014-17-1-2699 and the NSF under the grant NSF IIS-1830497.

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
