# OpenReview forum: "Planning Paths through Occlusions in Urban Environments"
_robot-learning.org/CoRL/2022/Conference — CoRL 2022 Oral_

### Official Review · Reviewer_hna9 · 2022-07-15

**Originality:** Good
**Technical Quality:** Good
**Clarity Of Presentation:** Very Good
**Impact:** 3

**Recommendation:**

Weak Accept: I recommend accepting the paper, but will not argue for my recommendation if the majority of other reviewers have a different opinion.

**Summary:**

This paper presents a new method for predicting occluded spaces.  The proposed inpainting model is then integrated with existing planning methods for navigation in occluded intersections and turns. The benefit of this approach compared to traditional methods that ignore map prediction is that it can design paths that are closer to the paths that the robot would have designed if it had access to the ground truth map. Additionally, the authors provide a dataset for training models of occluded turns and intersections for filling in unknown spaces. The proposed inpainting method employs the Pix2Pix network with a modified loss function while planning is achieved an existing planner, called hybrid A*.

**Issues:**

1) The authors should provide a more thorough literature review and discussing how this method compares against related works.
2) The benefit of the proposed method against planning methods that consider the unknown space as free should be better explained. For instance, the replanning frequency should be reported.
3) If space permits, this reviewer would encourage the authors to include figures where the proposed method does not predict an accurate map along with the designed paths.

**Quality Of The Limitations Section:**

Additional details required

**Reviewer Expertise:**

4: The reviewer is confident but not absolutely certain that the evaluation is correct

**Robotics Focus:**

Highly relevant to robotics but no hardware experiments

**Strengths And Weaknesses:**

Strengths:
1) A dataset for training models of occluded turns and intersections is proposed that can be quite useful for researchers in this area.

2) The proposed method performs better than methods that do not predict the unseen/occluded part of the environment in terms of how close the path is to the path that have been designed if the ground truth map was available.

Limitations:
1) As the authors admit, a limitation of the proposed method is that it assumes that during online deployment, the data come from the training data distribution. This, then, raises the question whether the proposed method can be really applied to unknown environments or not as it (since training data about the environment is available and at test time the system is applied in the same training environment). The authors should elaborate more on what "unknown environment" means.

2) Following up on the previous comment, the benefit/motivation of the proposed method should discussed in more details. Specifically, the above limitation implies that the proposed method can be applied only to known environments for which training data exists. If that's the case, why would an inpainting method be needed? Additionally, the authors should demonstrate more clearly the benefit of the their approach in planning. For instance, how does the (average) re-planning frequency of the proposed method compare to methods that do not predict the structure of occluded environments? In fact, to this reviewer's understanding, high-re-planning frequency is the main limitation of existing planning methods in unknown environments. Does the proposed method alleviate this? Also, it is worth noting that there exists a long list of works on navigating unknown environments without requiring (re)-planning. For instance  potential fields-based controllers for various tasks in unexplored semantic worlds have been proposed such as the following ones:

Arslan, O., & Koditschek, D. E. (2016, December). Sensor-based reactive navigation in convex sphere worlds. In The 12th International Workshop on Algorithmic Foundations of Robotics (WAFR).

Vasilopoulos, Vasileios, et al. "Reactive planning for mobile manipulation tasks in unexplored semantic environments." 2021 IEEE International Conference on Robotics and Automation (ICRA). IEEE, 2021.

What is the benefit of the proposed method against such works?

3) The literature review can be improved as there are several works focusing on predicting the structure of unseen environments; see e.g., the following papers and the references therein:
Georgakis, G., et al (2022). Uncertainty-driven planner for exploration and navigation. IEEE Conference on Robotics and Automation (ICRA), 2022

M. Narasimhan et al “Seeing the un-scene: Learning amodal semantic maps for room navigation,” in European Conference
on Computer Vision. Springer, 2020, pp. 513–529

4) Also, obviously, the proposed method cannot have 100% accuracy. It would be interesting to provide cases where e.g., the proposed method predicted an intersection which did not exist. How is planning affected by such wrong predictions?

**Summary Of Recommendation:**

Overall, the paper addresses a very interesting and important problem while the simulation studies show that it can outperform existing methods. This reviewer's major concern is about the contribution of the proposed algorithm as well as its practical importance compared to existing methods. As mentioned earlier, there are several methods that focus on predicting the unseen part of unknown environments. Also, the authors should elaborate more on the benefit of the proposed method with respect to planning and provide more detailed comparisons (see limitations 1-2).

---

> ### Author Response · Authors · 2022-08-24
> **Response to Reviewer hna9**
>
> **Comment:**
>
> Dear Reviewer, Thank you for your comments, here are our responses. Additional documents are attached in the ResponseToReviewers.zip file.
>
> Comment: A limitation of the proposed method is that it assumes that during online deployment, the data come from the training data distribution. Can the proposed method really be applied to unknown environments or not as it (since training data about the environment is available and at test time the system is applied in the same training environment). The authors should elaborate more on what "unknown environment" means.
>
> Response: We assume that for online navigation a previous map of the environment does not exist. We assume that the training and test data come from the same distribution (ie, both training and testing is for outdoor urban environments). For example, if our algorithm is trained on an urban environment, it would not be applied to a rural environment (see Paper Update pdf, page 1).
>
> Comment: The benefit/motivation of the proposed method should be discussed in more detail. The above limitation implies that the proposed method can be applied only to known environments for which training data exists. If that's the case, why is inpainting needed?
>
> Response: We assume the testing and training environments come from the same distribution. For example, if trained on an urban environment, the algorithm would only be applied to other urban datasets. We do not assume that the testing environments must be the exact same ones as from the training set. We do not test on the same roads and intersections the model is trained on.
>
> Comment: The authors should demonstrate more clearly the benefit of their approach in planning. How does re-planning frequency of the proposed method compare to methods that do not predict the structure of occluded environments? In fact, to this reviewer's understanding, high-re-planning frequency is the main limitation of existing planning methods in unknown environments. There exists a long list of works on navigating unknown environments without requiring (re)-planning. For instance potential fields-based controllers for various tasks in unexplored semantic worlds have been proposed....
>
> What is the benefit of the proposed method against such works?
>
> Response: Our focus is on the ability of the inpainting model for the planner to make more informed discrete high level decisions. For example, predicting a turn coming up at an intersection ahead versus not being able to see the turn due to occlusions and just planning straight ahead. Our framework gives the planner more high level options, especially in occluded environments that are common in congested areas and at long ranges. We do not focus on replanning, since the scope of our work is focused on the ability of our predictive model to discover navigable space in occluded regions.
>
> In Figure 3. (first column), we can see that without using predictive modeling, the baseline planner misses the turn (and intersection) entirely. Our model allows the planner to predict the occluded turns (Figure 3, third column), affording our planner the option of planning ahead for the turn.
>
> The referenced papers are focused on developing control schemes for collision avoidance and robot manipulation of objects in cluttered, unknown environments. To our understanding, these works do not have data-driven models to model the occluded areas of the scene and are focused on developing control schemes. Our work is based on using a data-driven model to predict semantic labels in occluded, unseen, spaces which allows for more informed high-level planning (planning "through" occluded spaces).
>
> Comment: The literature review can be improved as there are several works focusing on predicting the structure of unseen environments; see e.g., the following papers and the references therein....
>
> Response:  Thank you for suggesting these works, we have added them to our related works section (Paper Update pdf, pages 2 and 9).
>
> Comment: The proposed method cannot have 100% accuracy. It would be interesting to provide cases where e.g., the proposed method predicted an intersection which did not exist. How is planning affected by such wrong predictions?
>
> Response:
> While we do not have examples in the test dataset where intersections/turns are predicted where none exist, there are some examples where the predictive model does not predict an intersection/turn when one actually does exist. In these rare cases (Additional Evaluation pdf, Section 2), our planner is essentially providing the same information that the baseline planner believes so it behaves like the baseline planner and does not plan for the intersection/turn ahead of time.
>
> Comment: If space permits, this reviewer would encourage the authors to include figures where the proposed method does not predict an accurate map along with the designed paths.
>
> Response: We could not fit these into the main paper, but will add them into the supplemental material.
>
> **Zip File:**
>
> /attachment/9d06d83baeca753b616a794d8747491b7b9401dc.zip

---

> > ### Comment · Reviewer_hna9 · 2022-08-26
> > **Response to Authors**
> >
> > Thank you for your response which clarifies all my previous concerns. The contribution and the main goal of the paper is more clear now. As a minor comment, I would encourage you rephrase the first point in their list of contributions in Section 1. Specifically, instead of claiming that a new planning framework is provided, it would be better to say that a new inpainting model is proposed to fill in unknown spaces which can be incorporated into existing planning methods in unknown environments. Similarly, in Section 2, it would be better to highlight the "inpainting" term instead of "path planning". This way it will become more clear that the contribution of the paper is not on designing a new planner but on providing a method to fill unknown spaces which is indeed very useful for existing planners.

---

> > > ### Author Response · Authors · 2022-08-27
> > > **Response to reviewer hna9**
> > >
> > > Dear Reviewer,
> > >
> > > Thank you for reading our comments and responding. We appreciate your suggestions on clarifying the contributions.
> > >
> > > We have made edits to our paper and uploaded a pdf file below with highlights based on your feedback.

---

### Official Review · Reviewer_dgc4 · 2022-07-26

**Originality:** Good
**Technical Quality:** Good
**Clarity Of Presentation:** Very Good
**Impact:** 3

**Recommendation:**

Weak Accept: I recommend accepting the paper, but will not argue for my recommendation if the majority of other reviewers have a different opinion.

**Summary:**

In this paper the authors leverage an existing image generation neural network to fill in (or in-paint) missing and occluded drivable areas of the ground vehicle’s sensor map. The input is a bird’s eye view translation of a semantic LIDAR sensor. The neural net is trained with a custom loss function intended to increase in-painting area and preserve small features. The resulting image prediction is used as the basis for an occupancy map.

The authors use the Hybrid-A∗ algorithm on the prediction map to plan a trajectory that closely matches what would be generated if no occlusions were present.


**Issues:**

- It is not entirely clear to this reviewer what the main improvements are over [5] as comparisons are made against generic in-painting algorithms. The main difference as I understand is that they use a different GAN and that [5] requires an explicit labeling (i.e., a mask to remove the areas that do not need to be in-painted), and the use of Hybrid-A* instead of A*.  More discussion is needed on why this explicit labelling is such a significant drawback.

- The introduction raises the safety issue of planning as if space is free, but then does so. It is not clear how this approach is different/better than having a coarse map.

- The paper could use a stronger motivation for when no coarse map of the environment would be availablpe a priori.

- The comparison to Pix2Pix does not seem necessary as Pix2PixHD is an improvement on that.

******************************************************
Revised review:  Based on the reviewer responses I am updating my rating to weak accept and removing criticisms about the lack of consideration for dynamic obstacles
*******************************************************

**Quality Of The Limitations Section:**

Additional details required

**Reviewer Expertise:**

4: The reviewer is confident but not absolutely certain that the evaluation is correct

**Robotics Focus:**

Highly relevant to robotics but no hardware experiments

**Strengths And Weaknesses:**

Strengths:
- The authors illustrate nicely the ability of the trained system/GAN to approximate a local map from only LIDAR data

- The results when compared to Pix2PixHD and DeepFillV2 are impressive with clear improvements.

Weaknesses:
- There is no comparison of results with the paper that most closely resembles their work (i.e., ref [5] in the paper). Instead, they compare only against generic in-painting algorithms (Pix2Pix, Pix2PixHD, and DeepFillV2) then plan using their motion planner.

- In the introduction, the authors comment that the two current approaches to planning in an occluded environment are to either plan as if space is free, or plan only in the visible map. They comment that if planning as if space is free, then they have to consider what may be in the space beyond the occlusion, e.g., pedestrians. However, in this paper, the probable free space is constructed and then used for planning – without regard to what may be occupying that space.

- There is no discussion about the cases in which this method fails and is incorrect in its prediction of free space. This is very important when considering safety.


**Summary Of Recommendation:**

The paper is a nice demonstration of adapting an existing in-painting neural net to predict the static shape of occluded space. The innovations seem limited to customizing the loss function and using a dynamics-based motion planner. While the authors are aware of a very similar paper that uses similar technology, they don’t adequately develop their improvements over this existing work. Finally, while they raise the safety issue of planning as if space is free, their algorithm subsequently plans trajectories as if there are no dynamic agents (e.g., pedestrians) in the occlusions.

---

> ### Author Response · Authors · 2022-08-24
> **Response to Reviewer dgc4**
>
> **Comment:**
>
> Dear Reviewer, Thank you for your comments, here are our responses. Additional documents are attached in the ResponseToReviewers.zip file.
>
> Comment: There is no comparison of results with the paper that most closely resembles their work (i.e., ref [5] in the paper). Instead, they compare only against generic in-painting algorithms (Pix2Pix, Pix2PixHD, and DeepFillV2) with their planner.
>
> Response: In ref [5], a convex polygon is drawn around the sparse lidar point cloud. Then the unknown spaces inside the convex polygon are inpainted. The clear drawback of this is that areas outside of the convex polygon are completely ignored when in reality, the data-driven models may be able to generate hypotheses of which semantic labels occupy those areas. Our framework does not limit the inpainted areas by the geometry of the initial lidar scan and uses a completely data-driven model to determine which pixels the model is able to fill in.
>
> In Figure 3 of our paper, where the model fills in the turn with generated labels, the approach from [5] would be unable to do that since it is bounded by the geometry of the initial lidar points (with the convex polygon) so the turn region would be excluded from the inpainting.
>
> Comment: The authors comment that the two current approaches to planning in an occluded environment are to either plan as if space is free, or plan only in the visible map. They comment that if planning as if space is free, then they have to consider what may be in the space beyond the occlusion, e.g., pedestrians. In this paper, the probable free space is constructed and used for planning – without regard to what occupies that space.
>
> Response: To clarify, by “planning as if space is free”, we mean completely unknown space (occluded from sensor measurements) as we assume navigation in an unexplored environment where no map exists. We are focused on the general structure of the environment in the unknown spaces, focusing on labels such as road, sidewalk, buildings. We use our data-driven model to fill in unknown spaces (due to occluding obstacles) and plan accordingly. Our paper mainly focuses on high level decision making such as in Figure 3, where our framework is able to predict an upcoming turn, while the baseline planner does not. This gives our framework the ability to make better high level decisions by predicting turns and intersections earlier (Figure 3.)
>
> For dynamic obstacles in occluded regions such as pedestrians, we assume an additional low-level obstacle collision/avoidance module is used such as the one in the ROS navigation stack. The focus of our paper is not on dynamic obstacles and we instead predict the scene structure in the unknown spaces. We have updated our paper to make this more clear (pages 1 and 5).
>
> Comment: There is no discussion about the cases in which this method fails and is incorrect in its prediction of free space. This is very important when considering safety.
>
> Response: We include additional studies of failure cases where the planned trajectory significantly differs from the ground truth trajectory. As we are mostly focused on the structure of the road beyond occluding obstacles, our datasets and model perform well and there are no examples in the test set where the model predicts a turn/intersection where none exist. There are rare cases where our model does not predict a turn when a turn does in fact exist, however, in these cases our model is essentially the same as the baseline and simply does not provide additional information the baseline does not already have (Additional Evaluation pdf, Section 2).
>
> Comment: What the main improvements are over [5].
>
> Response: See comment 1.
>
> Comment: The introduction raises the safety issue of planning as if space is free, but then does so. How is this different/better than a coarse map.
>
> Response: See comment 2.
>
> Comment: How does this relate to having a prior map, or HD map of the environment?
>
> Response: Our paper addresses a different problem space where we assume the environment is previously unknown. We do not assume a previously generated HD map is available for navigation.
>
> Comment: The comparison to Pix2Pix does not seem necessary as Pix2PixHD is an improvement on that.
>
> Response: We agree that Pix2PixHD is an improvement of Pix2Pix. We provide Pix2Pix as the most basic baseline as one of the comparisons.
>
> Comment: The paper does not really address the main problem with occlusions – there may be dynamic actors in the occluded space that impact safety.
>
> In addition, the method is based on the current LiDAR scan, so if part of the vehicle is seen, and then not seen any longer due to occlusion, my understanding is that this information will be lost. This should be clarified.
>
> Response: See comment 2.
>
> Our method is based on the current lidar scan, so information outside of the current scan is not considered in the inpainting. This may in fact result in loss of information from previous frames.
>
>
> **Zip File:**
>
> /attachment/dc7569103b8c22d30eca18c614fe436b0bc97202.zip

---

> > ### Comment · Reviewer_dgc4 · 2022-08-26
> > **Thank you for the detailed response**
> >
> > Thanks for the detailed response; it clears up several aspects.  A few follow-on questions/comments:
> >
> > Re: Comparison with [5] – The method of [5] appears to be using a limited field of view lidar scan (possibly a reflection that they also use their technique with vision). If [5] used a 360 degree scan as is done in your paper, then this limitation disappears? In-painting at the limit of the lidar range seems to be of limited use. Of greater importance, how does your proposed method of in-painting with the improved loss function out-perform [5]?
> > –
> > Re: Dynamic Environment and Safety – We noticed that paragraph 2 on page 1 has also been changed to replace the reference to avoiding hidden dynamic objects with ‘dead ends’. With the other changes, it is clear this paper is focused on predictive path planner in unknown environments.
> > However, the point on applicability stands. Even in unknown environments, the planner requires a coarse map (note: not HD) to be able to set the goal and determine the global path for the local planner to follow. If your in-painting method is only responsible for finding a local path, how does it improve over using the known connectivity?
> > –
> > Re: Prediction Failures – Is the fact that there are so few failures to predict a turn, and no predicted false turns a limitation of the dataset (drawing training and test from the same domain)? This should be clarified. Is there a mitigation strategy for unlikely condition of a false turn or is that left to the secondary safety planner?

---

> > > ### Author Response · Authors · 2022-08-26
> > > **Response to Reviewer dgc4**
> > >
> > > **Comment:**
> > >
> > > Dear Reviewer,
> > >
> > > Thank you for taking the time to read and respond to our comments. Here is our response; an additional zip file has been attached with a comparison of [5] and our model.
> > >
> > > Comment: Comparison with [5] – The method of [5] appears to be using a limited field of view lidar scan (possibly a reflection that they also use their technique with vision). If [5] used a 360 degree scan as is done in your paper, then this limitation disappears? In-painting at the limit of the lidar range seems to be of limited use. Of greater importance, how does your proposed method of in-painting with the improved loss function out-perform [5]?
> > >
> > > Response: We provide an image example of the limitations of [5]. In (Polygon.png), taken from Figure 3 of our paper, the yellow polygon represents where [5], using the same 360 degree scan as our dataset, would limit the inpainting to; as it uses a convex polygon to decide where to inpaint (left). In contrast, our model is not restricted to manual annotations (yellow polygon) and is much more flexible. Using [5], the top turn inpainted (image left) would not be clear, while in our model, we are able to predict the turn clearly.
> > >
> > > To compare the performance, [5] claims an mean Intersection-over-Union (mIoU) of 13.10 (on their dataset). We calculate the mIoU of our model on our dataset and it is 60.15 (higher is better). However, it should be noted that the mIoU stated for [5] is only on the inpainting inside the convex polygon, while the one for our work is done for the entire sensor measurement, so it is not an exact comparison.
> > >
> > > We were unable to run the code from [5] on our dataset due to GPU incompatibilities.
> > >
> > > Comment: Dynamic Environment and Safety – We noticed that paragraph 2 on page 1 has also been changed to replace the reference to avoiding hidden dynamic objects with ‘dead ends’. With the other changes, it is clear this paper is focused on predictive path planner in unknown environments. However, the point on applicability stands. Even in unknown environments, the planner requires a coarse map (note: not HD) to be able to set the goal and determine the global path for the local planner to follow. If your in-painting method is only responsible for finding a local path, how does it improve over using the known connectivity?
> > >
> > > Response: We assume only knowledge of the global goal location relative to the current pose. In Figure 3 for example, we know the global goal is to the top right of the image. However, we do not assume any coarse map exists. The planners in our work attempt to plan the fastest route to the global path given the known and predicted space by taking a greedy approach and finding the closest waypoint in the road segmentation mask. The scope of our work is prediction of occluded structures and obstacles to allow for more informed path planning for the local planner. No previous coarse map information is assumed other than the global goal.
> > >
> > > Comment: Prediction Failures – Is the fact that there are so few failures to predict a turn, and no predicted false turns a limitation of the dataset (drawing training and test from the same domain)? This should be clarified. Is there a mitigation strategy for the unlikely condition of a false turn or is that left to the secondary safety planner?
> > >
> > > Response: The fact that there are few failures of predicting a turn could be because the dataset is not complex enough. However, the KITTI dataset and its derivatives are widely accepted as a baseline dataset in the self-driving community, and we believe it is fairly representative of navigation in an urban outdoor environment. As the dataset complexity increases, then the performance of the predictive model would naturally decrease.
> > >
> > > In the unlikely condition a false turn is predicted, a secondary safety planner would be tasked with stopping and replanning.
> > >
> > >
> > >
> > > **Zip File:**
> > >
> > > /attachment/c81ebffdb5c736b53b45b5ede299bfeb09773658.zip

---

### Official Review · Reviewer_apxG · 2022-07-31

**Originality:** Fair
**Technical Quality:** Fair
**Clarity Of Presentation:** Good
**Impact:** 3

**Recommendation:**

Weak Reject: I recommend rejecting the paper, but will not argue for my recommendation if the majority of other reviewers have a different opinion.

**Summary:**

In this paper, the authors propose a framework for motion planning under occlusion when driving through dense urban environments. The framework project LiDAR scans onto a BEV (bird eye view) map and fills in unknown regions of the map using a trained inpainting netowrk. The inpainting network is similar to Pix2pix HD, but the authors propose additional training losses to address issues with lost semantic labels from the original map, and missing small details. The framework then uses a hybrid A* planner to plan dynamically feasible paths through the inpainted map

**Issues:**

See weaknesses above.

**Quality Of The Limitations Section:**

Additional details required

**Reviewer Expertise:**

3: The reviewer is fairly confident that the evaluation is correct

**Robotics Focus:**

Highly relevant to robotics but no hardware experiments

**Strengths And Weaknesses:**

Strengths
- The paper addresses an important real world problem and evaluates the effectiveness of the proposed algorithm using the real world KITTI dataset
- The paper provides statistical results and an example scenario where the proposed framework clearly outperforms previous methods. However, I believe more analysis is needed (see below).

Weaknesses
1. In the introduction, the authors state that considering unknown spaces as free “increases the chances of varied maneuvering and collisions”. Safety should be considered one of the main motivations of the work. As such, I would like to see experimental results to include metrics that captures this, such as collision rate.
2. Similarly, the OL baseline seemed to consider unknown spaces as restricted. A comparison should also be done where unknown spaces are considered as free. This baseline will help motivate the work more.
3. The figures show clear improvement for a clear intersection. However, additional figures should be included to show predictions and planning results in congested intersections, which seem to be a more important use case.
4. The authors claim that the two additional losses improved the framework's performance. Statistical ablation results should be provided to determine exactly which loss contributes the most to performance of the planner.
5. This work seems to consider obstacles in the environment as static. In reality, other cars or pedestrians are not static. How does this framework perform in such scenarios? Is this a limitation of the algorithm?
6. One contribution mentioned in the introduction is a dataset for training occlusion aware models. However, the dataset used in the paper does not seem to be a new dataset, but simply a post processing of an existing dataset. (This is a weak criticism, and may be due to my misunderstanding)

**Summary Of Recommendation:**

I believe that this work addresses an important issue with autonomous driving. The authors provide some results showing scenarios where the algorithm outperform previous methods. However, I do not believe there is enough experimental results that justify that this algorithm is safer than previous methods or will perform better in more important scenarios such as when congestion is occluded. As such, I tentatively recommend this paper to be rejected unless the authors can provide results that addresses the weaknesses stated above.

---

> ### Author Response · Authors · 2022-08-24
> **Response to Reviewer apxG**
>
> **Comment:**
>
> Dear Reviewer,
> Thank you for your comments, here are our responses. Additional documents are attached in the ResponseToReviewers.zip file.
>
> Comment: Considering unknown spaces as free “increases the chances of varied maneuvering and collisions”. Safety should be considered one of the main motivations of the work. I would like to see experimental results to include metrics that captures this, such as collision rate.
>
> Response: Our framework is focused on a mid-level planner which generates a sequence of way points which a more low-level planner would follow. Specifically, we assume an obstacle/collision avoidance module such as the one in the ROS navigation stack would be used in addition to our planner for obstacle avoidance. The main motivation of our work is developing a framework for planning through unknown spaces, particularly those further away which cannot be seen via the sensors; we assume low-level safety behavior can be left to off-the-shelf obstacle avoidance algorithms, as that is not the focus of our work (Highlighted in the Paper Update pdf pages 1 and 5).
> We do provide additional analysis of when the predictive model predicts incorrect structures (Please refer to the Additional Evaluation pdf, Section 2).
>
> Comment: The OL baseline seemed to consider unknown spaces as restricted. A comparison should also be done where unknown spaces are considered as free. This baseline will help motivate the work more.
>
> Response: Our framework is focused on filling in unknown spaces with our data-driven model. The baseline we compare to is one which is unable to hypothesize about unknown spaces. For our work we are specifically focused on planners which are restricted by unknown space in order to demonstrate the merit of our predictive model on these unknown spaces by extending the navigable space. We believe this type of planner would be a complement to a collision avoidance path planner which only plans in known/viewable spaces.
>
> Comment: The figures show clear improvement for a clear intersection. However, additional figures should be included to show predictions and planning results in congested intersections, which seem to be a more important use case.
>
> Response: We provide additional simulations with parts of the original lidar scan (OL) masked out to simulate congested intersections and road conditions. The KITTI-360 dataset does not have many congested scenarios so we generate our own synthetic “congested” dataset by masking out additional sections of the sparse lidar scans. We provide additional studies and analysis of these congested road conditions that demonstrate the effectiveness of our framework even with additional synthetic occlusions (Please refer to the Additional Evaluation pdf, Section 1). Even with multiple simulated occlusions, our framework still outperforms the baseline (OL) across almost all metrics.
>
> Comment: The authors claim that the two additional losses improved the framework's performance. Statistical ablation results should be provided to determine exactly which loss contributes the most to performance of the planner.
>
> Response: Please refer to our supplemental page table A1 our ablation results on our two additional losses.
>
> Comment: This work seems to consider obstacles in the environment as static. In reality, other cars or pedestrians are not static. How does this framework perform in such scenarios? Is this a limitation of the algorithm?
>
> Response: Our algorithm does not explicitly consider dynamic obstacles in our model. The main focus of the paper is on modeling the occluded parts of the scene, specifically targeting the road structure which may be occluded due to obstacles in the scene - particularly at far ranges which are often occluded. Our algorithm simply fills in unseen pixels with our inpainting model based on the known pixels. Dynamic obstacles should not affect this process.
> As we are focused on predicting the underlying structure of the scene, we also assume our planner would be combined with an obstacle/collision module for avoidance of dynamic obstacles. Our paper has been updated to make this more clear (highlighted in Paper Update pdf, page 1 and page 5).
>
> Comment: One contribution mentioned in the introduction is a dataset for training occlusion aware models. However, the dataset used in the paper does not seem to be a new dataset, but simply a post processing of an existing dataset. (This is a weak criticism, and may be due to my misunderstanding)
>
> Response: The dataset introduced in the paper is indeed an augmentation of the KITTI-360 dataset. We specifically augment the KITTI-360 data for the planning through occluded environments task by first projecting semantic labels from the fully annotated point cloud (Ground Truth) back to the sparse lidar scans and transforming the points to a bird’s eye view (BEV) perspective. We provide matching BEV training pairs consisting of sparse lidar scans and the ground truth map.
>
> **Zip File:**
>
> /attachment/578f0c64a46f768833ad86c97745ef6a3d45e044.zip

---

> > ### Comment · Reviewer_apxG · 2022-08-27
> > **Response to authors**
> >
> > Dear Authors,
> >
> > Thank you for your response and for taking my suggestions into consideration. I think I understand the scope of the work a bit better now. Many of my prior comments came from an incorrect understanding that this method aimed to predict what pedestrians or vehicles are hidden by occlusion and navigate around them. Some comments in the paper about how this work fits into the full navigation stack could be helpful to readers

---

> > > ### Author Response · Authors · 2022-08-28
> > > **Response to reviewer apxG**
> > >
> > > Dear Reviewer,
> > >
> > > Thank you for reading our response and your additional suggestions.
> > >
> > > We have updated our paper with some comments about how our work fits into a full navigation stack in the attached pdf (highlighted page 5).

---

### Official Review · Reviewer_7Fav · 2022-08-03

**Originality:** Very Good
**Technical Quality:** Very Good
**Clarity Of Presentation:** Excellent
**Impact:** 3

**Recommendation:**

Strong Accept: I recommend accepting the paper and will argue for my recommendation even if other reviewers hold a different opinion.

**Summary:**

This paper presents a new path-planning framework for autonomous vehicle where the visibility is limited to the lidar sensor data. The authors provide a planner based on the hybrid A* algorithm, which plans kinematically feasible paths on 2D grids, with a neural network capable of adding new parts not the grid map based on statistical correlations.

The noticeable contribution with respect to previous work is to be able to do modify the map without requiring labeling of the pixels that need to be inpainted.

**Issues:**

None.

**Quality Of The Limitations Section:**

Limitations are addressed clearly

**Reviewer Expertise:**

4: The reviewer is confident but not absolutely certain that the evaluation is correct

**Robotics Focus:**

Highly relevant to robotics but no hardware experiments

**Strengths And Weaknesses:**

In the recent years, much of machine learning breakthroughs has been in the field of generative models and unsupervised learning. This work leverages state of the art machine learning for dealing with occlusions from sensors. Instead of using a traditional filtering approach the authors purely resort to treating the filtering problem in output space by inpating the grid map’s pixels using a generative model. They make use of computer vision state of the art models to perform this inpainting.

One of the strength of the paper is to provide dedicated training losses to the problem of inpainting for occlusions. Particularly they introduce reconstruction losses that take care of the zero pixels in the generated images explicitly and a loss that deals with local patches. These modifications quantitatively ameliorate the prediction.

The paper compares to state of the art inpainting prediction network and is shows to outperform these methods.
The authors outline limitations especially concerning the fact that the inpainting behavior is limited to the data and that it relies on exact localization of the agent.

These make sense and while it would be nice to see extension of the work with other filtering mechanisms to handle uncertainty and data transfer approaches to deal with the data distribution.  The paper seems to adequately represent the state of the art.


**Summary Of Recommendation:**

This paper presents a very sound contribution, however potentially incremental. It takes place in a very interesting research avenue, where data driven approaches clearly make sense.

---

> ### Author Response · Authors · 2022-08-24
> **Response to Reviewer 7Fav**
>
> **Comment:**
>
> Dear Reviewer,
>
> Thank you for your comments. We are glad you enjoyed reading our paper!
>
> We agree that adding uncertainty analysis to our framework would be a good future extension.
>
> We have attached a zip folder with some minor edits and additional evaluation.
>
> **Zip File:**
>
> /attachment/56b41fcf7faaff549101db86691ad7597df93955.zip

---

### Meta-Review · Area_Chair_yb99 · 2022-08-09

**Recommendation:** Accept (Oral)
**Confidence:** 5

**Metareview:**

Summary
The authors present a framework for motion planning in congested urban areas. The framework project LiDAR scans a BEV map and fills in unknown parts using a trained inpainting network. The inpainting network is comparable to Pix2pix HD.
The proposed inpainting model is then combined with existing navigation planning methods. This strategy can create paths closer to what the robot would have designed with the ground truth map than standard methods that disregard map prediction.
The authors propose extra training losses to address missing semantic labels and minor details. A hybrid A* planner plans dynamically feasible paths through the inpainted map.


Strengths
1. A dataset for training models of occluded turns and junctions is proposed.
2. The suggested strategy outperforms methods that do not forecast the unseen/occluded component of the environment.
3. Results are better than Pix2PixHD and DeepFillV2.
4. The research tests the suggested technique using the real-world KITTI dataset.
5. The paper offers statistical findings and a situation where the suggested framework outperforms existing methods. More research is needed.
6. Recent machine learning advancement has been in generative models and unsupervised learning.
7. This work uses machine learning to handle sensor occlusions. Instead of a standard filtering strategy, the authors inpaint grid map pixels with a generative model. Inpainting uses cutting-edge computer vision models.
8. The paper provides separate training losses for occlusion inpainting. They introduce reconstruction losses for zero pixels and local patches. This improves prediction quantitatively.
9. The paper outperforms state-of-the-art inpainting prediction networks. The authors note that inpainting is restricted to data and requires precise agent location.

Update:
The concerns raised have been clarified and the revised submission substantially updated.

Weaknesses
1. The authors say that interpreting unknown locations as free "increases maneuvering and collisions." Safety should be a top priority. I'd want to see experimental findings that include collision rate.
2. OL seems to restrict unknown regions. Unknown spaces should also be compared.
3. Figures show an improved junction. Additional data should show predictions and planning results in congested intersections, a more important use case.
4. Two additional losses improved the framework's performance, say the authors. Statistical ablation findings should show which loss contributes more to planner performance.
5. This work considers environmental obstacles static. Cars and pedestrians aren't static. How does this framework perform? Is this a programming error?
6. A dataset for training occlusion-aware models was given in the introduction. The paper's dataset appears to be a post-processing of an existing dataset. (This is a weak criticism owing to my misunderstanding.)
7. No comparison of findings with similar paper (i.e., ref [5] in the paper). Instead of comparing to generic in-painting algorithms, they use their motion planner.
8. The two current approaches to planning in an occluded environment are to plan as if space is free or to plan just in the viewable map. They note that if they design as if there is no occlusion, they must include pedestrians. In this paper, however, the expected open space is built and then used for planning.
9. It's not clear when this algorithm anticipates empty space wrongly. Safety requires this.
10. The authors admit that online deployment presupposes training data is used. The proposed method's applicability to unknown contexts is questioned (since training data about the environment is available and at test time the system is applied in the same training environment). "Unknown environment" needs clarification.
11. The proposed method's benefit/motivation should be addressed in more detail.


**Best Paper Nomination:**

No